# Association of α-Dicarbonyls and Advanced Glycation End Products with Insulin Resistance in Non-Diabetic Young Subjects: A Case-Control Study

**DOI:** 10.3390/nu14224929

**Published:** 2022-11-21

**Authors:** Melinda Csongová, Jean L. J. M. Scheijen, Marjo P. H. van de Waarenburg, Radana Gurecká, Ivana Koborová, Tamás Tábi, Éva Szökö, Casper G. Schalkwijk, Katarína Šebeková

**Affiliations:** 1Institute of Molecular Biomedicine, Medical Faculty, Comenius University, 811 07 Bratislava, Slovakia; 2Department of Internal Medicine, Maastricht University Medical Centre, 6200 MD Maastricht, The Netherlands; 3Institute of Medical Physics, Biophysics, Informatics and Telemedicine, Faculty of Medicine, Comenius University, 813 72 Bratislava, Slovakia; 4Department of Pharmacodynamics, Faculty of Pharmacy, Semmelweis University, 1089 Budapest, Hungary

**Keywords:** advanced glycation end products, α-dicarbonyls, D-lactate, cardiometabolic risk, insulin resistance, sex differences, sRAGE, sVAP-1

## Abstract

α-Dicarbonyls and advanced glycation end products (AGEs) may contribute to the pathogenesis of insulin resistance by a variety of mechanisms. To investigate whether young insulin-resistant subjects present markers of increased dicarbonyl stress, we determined serum α-dicarbonyls-methylglyoxal, glyoxal, 3-deoxyglucosone; their derived free- and protein-bound, and urinary AGEs using the UPLC/MS-MS method; soluble receptors for AGEs (sRAGE), and cardiometabolic risk markers in 142 (49% females) insulin resistant (Quantitative Insulin Sensitivity Check Index (QUICKI) ≤ 0.319) and 167 (47% females) age-, and waist-to-height ratio-matched insulin-sensitive controls aged 16-to-22 years. The between-group comparison was performed using the two-factor (sex, presence/absence of insulin resistance) analysis of variance; multiple regression via the orthogonal projection to latent structures model. In comparison with their insulin-sensitive peers, young healthy insulin-resistant individuals without diabetes manifest alterations throughout the α-dicarbonyls-AGEs-sRAGE axis, dominated by higher 3-deoxyglucosone levels. Variables of α-dicarbonyls-AGEs-sRAGE axis were associated with insulin sensitivity independently from cardiometabolic risk markers, and sex-specifically. Cleaved RAGE associates with QUICKI only in males; while multiple α-dicarbonyls and AGEs independently associate with QUICKI particularly in females, who displayed a more advantageous cardiometabolic profile compared with males. Further studies are needed to elucidate whether interventions alleviating dicarbonyl stress ameliorate insulin resistance.

## 1. Introduction

Insulin resistance is a pathological condition in which the physiological response to insulin stimulation of target tissues such as the liver, the skeletal muscle, and the adipose tissue, is impaired. To maintain glucose uptake and utilization, the organism increases insulin production, leading to hyperinsulinemia. Due to the pleiotropic effects of insulin, insulin resistance co-occurs with other cardiometabolic abnormalities, such as visceral adiposity, hypertension, dyslipidemia, hyperuricemia, and low-grade inflammation. As a consequence, insulin resistance represents an increased risk to develop type 2 diabetes [1].

Advanced glycation end-products (AGEs) are a family of non-enzymatically glycated molecules formed as a part of normal metabolism via several chemical routes (Appendix A) [2,3,4,5]. Reactive α-dicarbonyls such as methylglyoxal (MGO), glyoxal (GO), or 3-deoxyglucosone (3-DG) are major precursors of AGEs [6,7]. AGE-modification alters the structure and function of proteins. In addition, the interaction of AGEs with cell-surface receptors for AGEs (RAGE) activates downstream pathways promoting oxidative stress, inflammation, and atherogenesis [8,9]. The generation of AGEs is accelerated under conditions of chronic hyperglycemia and the process has been implicated in the pathogenesis of several chronic degenerative diseases, including diabetes [10,11].

Recent data suggest that AGEs and reactive α-dicarbonyls may contribute to the pathogenesis of insulin resistance by different underlying mechanisms. Glycation of insulin impairs its action [12,13]. AGEs induce insulin resistance by interfering with the insulin signaling pathways through the RAGE-dependent induction of pro-inflammatory cytokines and reactive oxygen species (ROS) [14,15]. MGO or MGO-modified proteins affect ß-cell function, interfere with cellular insulin signaling independently of intracellular ROS formation, and induce insulin resistance via modification of AMP-kinase (reviewed in [16]). A 3-DG may induce hepatic insulin resistance by decreasing the insulin-induced expression of GLUT2, the insulin-stimulated tyrosine phosphorylation of IRS-1, and the expression of the downstream target proteins [17].

Studies in rodents and clinical observations indicate an association between α-dicarbonyls, AGEs, and insulin resistance. In obese insulin-resistant (IR) mice, serum AGE levels positively correlated with insulinemia [18]. MGO administered to rats induced insulin resistance accompanied by an increase in circulating N^ε^-(carboxyethyl)lysine (CEL) [19]. Insulin-resistant adults without diabetes displayed higher levels of circulating 3-DG [20], AGEs [21,22], and lower concentrations of soluble RAGE (sRAGE) [23,24] compared with their insulin-sensitive (IS) peers. 

Whether insulin resistance is associated with alterations in the α-dicarbonyls-AGEs-RAGE axis already at a young age, remains unclear. To clarify this issue, we studied the association of serum levels of α-dicarbonyls, free- and protein-bound (pb) AGEs, urinary AGEs, and circulating sRAGE concentrations in IS and IR young healthy subjects. We anticipated that even in individuals free from comorbidities, dicarbonyl stress is associated with insulin resistance, resulting in the accumulation of circulating AGEs, compensated by their higher urinary excretion and a decline in sRAGE levels. 

## 2. Subjects and Methods

### 2.1. Study Design and Population

This is a subanalysis of data obtained in the cross-sectional study Respect for Health, conducted in the school year 2011/2012. The survey aimed to assess the health status of students attending state-governed secondary schools in the Bratislava Region of Slovakia. The study protocol was approved by the Ethics Committee of the Bratislava Self-governing Region. Written informed consent was obtained from full-aged participants; in minors, their verbal assent and consent signed by parents or caregivers were acquired. Participation was voluntary. Exclusion criteria were any acute or chronic illness, pregnancy, or lactation in females. 

The study design was described in detail previously [25]. Briefly, 2769 White Caucasians of Central European descent aged 11–32 years participated in the study. For the current analysis, we extracted data from 16-to-22-year-old individuals without diabetes (*n* = 2357) in whom insulin sensitivity was assessed using the Quantitative Insulin Sensitivity Check Index (QUICKI) [26]. There neither are recommendations from professional societies regarding the threshold values of surrogate markers to define insulin resistance nor a consensus on the percentile criterion for the cutoffs [27]. In this study, insulin resistance was set arbitrarily at the 15th percentile, i.e., QUICKI ≤ 0.319. From IR subjects, we selected those in whom all data on anthropometry, blood pressure (BP), and blood chemistry were available and there was sufficient biological material for the analyses of selected α-dicarbonyls and AGEs using the ultra-performance liquid chromatography-tandem mass spectrometry (UPLC-MS/MS) [28,29] method (*n* = 142). One-hundred-and-sixty-seven sex-, age-, and waist-to-height ratio (WHtR)-matched IS subjects served as controls.

### 2.2. Measurements

Anthropometric and BP measurements were performed according to standard protocols by medical personnel. Height was measured using a portable stadiometer; waist circumference with non-elastic tape; body weight, and total body fat (TBF) percentage using digital scales (Omron BF510, Kyoto, Japan). Body mass index (BMI) and WHtR were calculated.

In a subject seated for 10 min, BP was measured using a digital monitor (Omron M-6 Comfort, Kyoto, Japan). An average of the last two measurements out of three was recorded.

Blood and spot urine samples were collected at health centers between 7.00–9.00 a.m., after overnight fasting. In the central laboratory, sera were analyzed for glucose, total cholesterol, high-density lipoprotein cholesterol (HDL-C), triacylglycerols (TAG), creatinine, uric acid, high-sensitive C-reactive protein (CRP), and insulin using standard laboratory methods (Advia 2400 analyzer, Siemens, Munich, Germany). Urinary albumin (turbidimetrically) and creatinine were determined. Albumin-to-creatinine ratio (ACR), estimated glomerular filtration rate (eGFR, using the equation for the full age spectrum, with Q-value matching height) [30], and non-high-density lipoprotein cholesterol concentrations (=total cholesterol minus HDL-C) were calculated. The cardiometabolic risk was estimated using a continuous metabolic syndrome score (cMSS = WHtR/0.5 + systolic BP/130 + glycemia/5.6 + TAG/1.7 − HDL-C/1.02 (males) or 1.28 (females)) [31]. A 4-component score (cMSS4, e.g., without glycemia) was used for multivariate analyses.

At the Institute of Molecular Biomedicine, soluble receptor for advanced glycation end-products (sRAGE, R&D Systems Inc., Minneapolis, MN, USA, which determines the total pool of all soluble forms of RAGE, including endogenous secretory RAGE-esRAGE), esRAGE (Daiichi Fine Chemicals Co. Ltd., Takaoka, Japan), and soluble vascular adhesion protein-1 (sVAP-1, eBioscience, Vienna, Austria) were analyzed using commercial ELISA kits, according to the manufacturer’s instructions. The concentration of cleaved RAGE (cRAGE) was calculated as sRAGE minus esRAGE; the cRAGE-to-esRAGE ratio (esRAGE:cRAGE) was expressed in arbitrary units. Serum fructosamine [32] and urinary D-lactate (Abcam kit, Cambridge, UK; for insufficient sample amount, analyzed only in 68% of males and 63% of females) were determined spectrophotometrically, and serum thiobarbituric acid reactive substances (TBARS) fluorometrically [33], using the Sapphire II instrument (Tecan, Vienna, Austria). The d-lactate-to-creatinine ratio was calculated. In aliquots of samples transferred on dry ice, serum activity of semicarbazide-sensitive amine oxidase (SSAO, EC 1.4.3.21) was determined radiometrically by liquid scintillation counting [34] in Budapest (only in males, *n* = 119, insulin-sensitive: 52.9%), and expressed as nmol of benzaldehyde formed by 1 mg of serum protein in 1 h at 37 °C; and at Maastricht University, the UPLC-MS/MS method was used to analyze serum concentrations of MGO, GO, 3-DG; free MG-H1, CML, CEL; and protein-bound pentosidine, MG-H1, CML, and CEL, as published in detail previously [28,29]. Urinary concentrations of MG-H1, CML, and CEL were due to small sample volumes not determined in three males (two IS) and sixteen females (eight IS). Urinary AGEs were normalized to urinary creatinine. Fractional excretion (FE) of free-AGEs was calculated as the ratio of the clearance of free-AGE to that of creatinine. Assuming that creatinine is excreted solely by glomerular filtration, FE < 1 indicates tubular reabsorption of the substance of interest, while FE > 1 points to its tubular secretion.

### 2.3. Sample Size Calculation

Post-hoc power calculation on means and standard deviations (SD) of QUICKI in IS vs. IR groups indicated a power of 100% at α = 0.001, both in males and females. As to the estimation of the sample size for multivariate analysis, approaches prioritizing minimal total sample size recommend numbers from 50 to 400; concepts based on the ratio of participants to variables suggest a minimum participant-to-item ratio ranging from 5:1 to 10:1 [35]. In the current study, the ratio was 7.7:1 for males and 7.0:1 for females. Thus, the numbers of our probands are within the ranges required by both approaches.

### 2.4. Statistical Analyses

Data not fitting the Gaussian distribution (Shapiro-Wilk normality test) were logarithmically (log) transformed. The impact of sex, the presence or absence of insulin resistance, and their interaction with dependent variables were tested using the two-factor analysis of variance (ANOVA). Pearson’s correlation coefficients were calculated. Categorical data were compared via the Chi-square test. Data are presented as mean ± SD, back-transformed log data as geometric mean (−1 SD, +1 SD), and categorical data as counts and percentages. The *p* values of <0.05 were considered statistically significant. Analyses were performed using GraphPad Prism 6.0 (GraphPad Software, San Diego, CA, USA), and SPSS software (v.16 for Windows, SPSS, Chicago, IL, USA).

Multivariate regression using the Orthogonal projections to the latent structures (OPLS) model was employed to identify the set of independent variables predicting the QUICKI (Simca v.17 software, Sartorius Stedim Data Analytics AB, Umea, Sweden). The variance inflation factor was used to check for the problem of multicollinearity in multiple regression analysis. Regression on the QUICKI (outcome variable) was performed by entering analyzed α-dicarbonyls, free-, protein-bound-, and urinary AGEs, sVAP-1, cRAGE, CMSS4, fructosamine, TBF, non-HDL-C, CRP, and TBARS as independent determinants. Before modeling, variables with high skewness and low min/max ratio were log-transformed and all data were mean-centered. Variables with Variables important for the projection (VIP) values ≥1.0 were considered significant predictors.

## 3. Results

The general characteristics of the study population stratified by sex and insulin sensitivity are shown in Table 1.

### 3.1. Insulin Sensitivity

The prevalence of IS and IR individuals was similar among males and females (Table 1). Insulin-resistant subjects presented higher fasting glucose, insulin, and fructosamine levels compared with their IS peers. Males and females displayed similar mean QUICKI values.

### 3.2. Cardiometabolic Risk Factors and Markers

Four groups (IS males, IR males, IS females, and IR females) displayed similar mean WHtR, total body fat percentage, and BP (Table 1). Compared with their IS counterparts, IR groups had higher BMI, non-HDL-C, and TAG levels; and males had lower HDL-C concentrations. Both IR groups displayed higher cardiometabolic risk estimated as cMSS, even after the exclusion of the glucose (cMSS4). Sex and the presence of IR independently affected CRP and uric acid levels; while the ANOVA indicated an interaction of both for TBARS. Females displayed slightly higher eGFR and urinary ACR compared with males.

### 3.3. Reactive Dicarbonyls and D-Lactate

Serum concentrations of MGO, GO, and 3-DG were higher in males compared with females (Table 2). Compared with IS subjects, 3-DG levels were higher in IR groups of both sexes, while only IR females presented higher GO concentrations than IS ones. Urinary excretion of D-lactate was similar in IS and IR groups; and higher in females than males. The correlation between log D-lactate and log serum MGO was significant in females (r = 0.212, *p* = 0.042) but not in males (r = −0.004, *p* = 0.963).

### 3.4. Free AGE-Adducts and Their Excretion

Serum levels of free-MG-H1, free-CEL, and free-CML were higher and urinary excretion of MG-H1, CEL, and CML was lower in males compared with females (Table 2). In both sexes, IR was associated with higher serum levels of free-MG-H1, free-CEL; and their higher urinary excretion. FE_MG-H1_ and FE_CML_ were higher in females compared with males; IR subjects displayed higher FE_CML_ than IS ones, and FE_CEL_ was similar across four groups of probands. Fractional excretions of free-AGEs showed an inverse relationship with their serum levels (males: r: −0.223 to −0.465, *p*: < 0.005 to <0.001; females: r: −0.168 to −0.233, *p*: < 0.034 to <0.008); and a direct correlation with urinary excretion (males: r: 0.349 to 0.468; females: 0.466 to 0.724, *p* < 0.001, all). The prevalence of FE_MG-H1_ > 1 reached 88.5% in IS and 80.6% in IR males (p_Chi_ = 0.163); and 88.6% and 88.1%, respectively, in females (p_Chi_ = 1.000). Eight percent of IS and 12.0% of IR males (p_Chi_ = 0.353); and 10.0% and 8.2%, respectively of females (p_Chi_ = 0.722), presented FE_CEL_ > 1. The prevalence of FE_CML_ > 1 reached 25.3% in IS and 40.1% in IR males (p_Chi_ = 0.044); and 38.6% and 49.2%, (p_Chi_ = 0.222), respectively, in females.

### 3.5. Protein-Bound AGEs

Males presented lower pb-CML and pb-CEL concentrations compared with females. Insulin resistance was associated with lower pb-MG-H1 and higher pb-pentosidine levels in males; while IR females presented higher pb-CEL levels compared with their IS peers (Table 2).

### 3.6. sRAGE, sVAP-1 Levels, and SSAO Activity

EsRAGE concentrations were higher in males compared with females but were not affected significantly by the presence of insulin resistance (Table 2). Insulin-resistant males but not females displayed higher total sRAGE, cRAGE levels, and cRAGE-to-esRAGE compared with their IS peers, confirming the disproportional rise of RAGE variants.

Soluble VAP-1 levels were higher in IR compared with IS groups (Table 2). There was a direct relationship between log sVAP-1 and glucose concentrations (males: r = 0.205, *p* = 0.009; females: r = 0.236, *p* = 0.004). In males, the activity of SSAO reached 60.0 ± 17.3 nmol mg^−1^ h^−1^; and showed a direct significant correlation with log sVAP-1 (r = 0.480, *p* < 0.001) but not with log MGO or MGO-derived AGEs (Appendix A).

### 3.7. Simple and Multivariate Regression of Independent Variables on QUICKI

There was a significant inverse relationship between 3-DG, free MGH-1, cMSS4, TBF, CRP, and QUICKI in both sexes (Table 3). In males, cRAGE, fructosamine, and non-HDL-C levels showed a negative, while pb-MGH-1 and pb-CML had a positive significant association with QUICKI. In females, serum concentrations of MGO, GO, free CEL, pb-CEL, and urinary excretion of MGH-1 and CEL inversely and significantly correlated with QUICKI; while sRAGE and cRAGE were associated with QUICKI directly (Table 3).

The OPLS model selected 3-DG, pb-MG-H1, and cRAGE (VIP: 1.86-to-1.04), and metabolic markers (cMSS4, fructosamine, TBF, and non-HDL-C; VIP: 1.44–1.12) as significant independent predictors of QUICKI in males (Table 3, Appendix A). The model explained 46% of the variation (R^2^) of QUICKI. In females, 3-DG, urinary MG-H1, urinary MGH-1, pb-CEL, free MGH-1, free CEL, and GO (VIP: 1.99-to-1.08); cMSS4 and TBF (VIP: 1.39, 1.09, respectively; R^2^: 48%; Table 3, Appendix A) significantly correlated with QUICKI in the OPLS model.

## 4. Discussion

Our data document that in young healthy individuals of both sexes, IR is associated with higher serum 3-DG levels; higher free-MG-H1 and free-CEL concentrations, and higher urinary excretions. This indirectly points to the higher production of MGO albeit the flux of MGO through the glyoxalase system seems not to be increased. Moreover, insulin resistance is associated with mildly higher pb-pentosidine concentrations in both sexes; higher pb-CEL levels in females; lower pb-MG-H1, and higher cRAGE concentrations in males (Appendix A). In multivariate regression, α-dicarbonyls and AGEs were associated more frequently with QUICKI in females, who displayed a more advantageous cardiometabolic profile compared with males. These data support the concept that derangement of the α-dicarbonyls-AGEs-sRAGE axis plays a pathogenetic role in the development of insulin resistance.

Endogenous α-dicarbonyls are intermediates of metabolic pathways of carbohydrates, lipids, and proteins. Observed sex differences in α-dicarbonyl concentrations could reflect mildly lower glycemia in females.

Experimental studies document that MGO contributes to the pathogenesis of insulin resistance via a modification of insulin structure and function, modulation of insulin secretion, and insulin signaling [13,16]. Its role in the induction of IR in humans has been widely disputed [16] but studies in subjects without diabetes are scarce. Obese IR females [36] but not males [37] displayed higher circulating MGO levels compared with their lean IS counterparts. Our IS and IR subjects presented similar serum MGO concentrations, although concentrations of sVAP-1 were higher in IR groups. With increasing glycemia, sVAP-1 levels increase to stimulate glucose uptake into muscle and liver cells [38]. Via its SSAO activity, sVAP-1 converts aminoacetone into MGO [39]. Despite the direct correlation between sVAP-1 protein mass and SSAO activity [40], no significant relationship between SSAO activity and MGO levels was observed. This reflects the observations that in contrast other amino acids, there is no significant association between threonine concentrations and insulin resistance *per se* [41,42]. Under a stable supply of threonine, aminoacetone represents only a minor source of MGO.

MGO is rapidly detoxified by the glyoxalase system into D-lactate [3]. Our data suggest that the flux of MGO through the glyoxalase system is higher in females compared with males. This is in line with the observation that tissue glyoxalase activity is also higher in females [43]. Similar D-lactate urinary excretion in our IS and IR subjects fits the data of Gugliucci et al. [44], that in adolescents, obesity rather than insulin resistance is associated with enhanced MGO degradation via the glyoxalase system. Yet, our data on free-MG-H1 and free-CEL levels and their urinary excretion indirectly point to a higher production of MGO in IR subjects. Whether MGO is excessively produced and bypasses the glyoxalase system in insulin-resistant states, remains a challenge for further studies.

Among the three investigated α-dicarbonyls, only 3-DG was increased in IR subjects; and in multivariate analyses, it was the most important predictor of insulin resistance in both sexes. Higher 3-DG levels were associated with higher insulin resistance also in healthy adults [20,45]. Experimental studies corroborate that 3-DG induces insulin resistance [17,46].

Higher free-AGEs concentrations in males are in line with higher levels of α-dicarbonyls and lower renal excretion of free-AGEs compared with females. Higher GO levels in IR females were not associated with increased f-CML levels or higher renal excretion of CML, probably reflecting that CML is also produced via several other metabolic pathways. While we observed higher free-MG-H1 and free-CEL concentrations in both IR groups; adult lean IS and obese IR males displayed similar levels of free-AGEs [37]. Thus, obesity might be an important factor in modifying circulating levels of free-AGEs in an insulin-resistant state.

Kidneys play a key role in the disposal of free-AGEs: they are filtered in glomeruli, reabsorbed and degraded in the proximal tubules, and thereafter excreted [47]. Most of our subjects eliminated free-MG-H1 via glomerular filtration and tubular secretion; while free-CEL and free-CML were predominantly filtered in glomeruli and thereafter reabsorbed in proximal tubules. Insulin resistance was associated only with mildly but significantly higher FE_CML_. Data from patients with type 1 diabetes [48] and healthy subjects with obesity [49] underline the high plasticity of the kidney in managing the flux of distinct free-AGEs differently, as well as their different handling in response to the severity and duration of hyperglycemia.

AGE-modified proteins are used as biomarkers of disease severity, its progression, risk of cardiovascular events, and mortality [50,51,52,53]. In adults, the association between pb-AGEs and insulin sensitivity is equivocal. Insulin resistance was positively associated with serum AGE levels determined using polyclonal antibody [21,22]; its relationship to pb-CML levels was either insignificant [22], or showed a significant direct relationship [54]; while pb-MGO-derived AGEs did not show an insignificant correlation with insulin resistance [54]. As to CML, individuals with obesity present lower levels of pb-CML compared with their lean peers; reflecting a RAGE-dependent accumulation of CML in adipose tissue where it induces inflammation and dysregulation of adipokines, thereby contributing to the development of obesity-associated insulin resistance [55]. Our IS and IR groups displayed similar pb-CML levels, probably reflecting their similar proxy measures of central obesity and total body fat percentage. IR subjects showed a mild elevation in minor pb-AGE—pentosidine, without a significant relationship to insulin resistance. However, its inverse relationship to QUICKI has been reported in healthy juveniles [56]. As to Pb-CEL and pb-MG-H1 serum levels, we revealed both sex- and insulin resistance-associated differences. Dissimilarities in pb-AGEs cannot be attributed to sex differences in protein turnover rate, as it does not differ between young males and females, despite the sexual dimorphism in body composition [57]. Our data show that the relationship between protein-bound AGEs and insulin sensitivity is not uniform, and the finding that insulin sensitivity is associated with higher pb-MG-H1 levels in males requires further verification.

SRAGEs arise either from proteolytic shedding of the RAGE receptor ectodomain (cleaved RAGE, cRAGE) or from alternative splicing of the RAGE gene (endogenous secretory RAGE-esRAGE) [58]. As the interaction of pb-AGEs with RAGE elicits downstream signaling [59], sRAGEs may act as decoys competitively inhibiting the binding of ligands to membrane-bound RAGE; thus, ameliorating the response towards tissue destruction [60]. Insulin-resistant adults display lower sRAGEs or esRAGE levels compared with their IS counterparts [23,24,61] and higher cRAGE levels and cRAGE:esRAGE were independently associated with decreased proportional odds for progression across the glucose tolerance continuum to type 2 diabetes [62]. A negative correlation between sRAGE or esRAGE and insulin resistance was reported for healthy children and adolescents [63,64,65]. As our IR males showed higher cRAGE concentrations and cRAGE:esRAGE compared with their peers, a question arises whether high cRAGE might be protective in the progression of insulin resistance. In line with the findings of Miranda et al. [62], our data suggests that cRAGE and esRAGE might be generated independently from each other. Insulin increases alternative splicing and secretion of esRAGE, as well as shedding of sRAGE [66,67]. However, these mechanisms do not elucidate the observed insulin resistance-associated sex differences in sRAGE levels.

In multivariate analyses, cardiometabolic risk factors were not the most important determinants of insulin resistance; probably because our probands displayed low cardiometabolic risk. In line with this assumption, α-dicarbonyls and AGEs were associated with insulin resistance more frequently and more strongly in females, who displayed a more advantageous cardiometabolic profile compared with males. Worsening of cardiometabolic health could eventually reverse the relevance of variables characterizing the α-dicarbonyls-AGEs-sRAGE axis vs. traditional cardiometabolic risk factors to insulin sensitivity. To prove or reject this hypothesis, longitudinal studies are needed.

The strengths of our study comprise an appropriate sample size; inclusion of young subjects suitable to explore changes occurring “early” given to onset of insulin resistance; determination of α-oxoaldehydes and AGEs concentrations in sera and urine using the “gold standard” UPLC-MS/MS technique; and simultaneous analyses of markers of α-dicarbonyls-AGEs-sRAGE axis and cardiometabolic risk factors. Limitations follow from the fact that the cross-sectional study design does not allow for evidence of causality. Insulin sensitivity was estimated using a surrogate marker—QUICKI. Determination of glyoxalase I activity could give additional information on the flux of MGO through the glyoxalase system. Reported associations could have also been affected by other factors not investigated herein, such as genetic background, family history, or lifestyle patterns. There is a limitation to generalizing our findings to populations with different epidemiological, anthropometric, or clinical characteristics.

## 5. Conclusions

Dicarbonyl stress might be an independent driver of insulin resistance already in young subjects, indicating the potential for α-dicarbonyls and AGEs-lowering interventions to prevent or ameliorate insulin resistance and the development of diabetes.

## Figures and Tables

**Table 1 nutrients-14-04929-t001:** Cohort characteristics.

	Males (*n* = 162, 52.4%)		Females (*n* = 147, 47.6%)		Two-Factor ANOVA (*p*)		
	IS (28.8%)	IR (23.6%)	IS (25.2%)	IR (22.3%)	Sex	IS/IR	Sex × IS/IR
***n*, %**	89 (54.9)	73 (45.1)	78 (53.1)	69 (46.9)	0.741 ^Chi^		
**Age, years**	18.1 ± 0.90	17.6 ± 0.7	17.4 ± 1.2	17.6 ± 1.4	**0.001**	0.323	**0.010**
**Glucose, mmol/L**	4.8 ± 0.3	5.2 ± 0.5	4.6 ± 0.4	5.0 ± 0.5	**<0.001**	**<0.001**	0.080
**FA, mmol/L**	1.7 ± 0.5	2.0 ± 0.6	1.6 ± 0.3	1.7 ± 0.4	**0.001**	**0.001**	0.080
**Insulin, µIU/mL**	8.1 (5.6, 11.7)	21.3 (14.5, 31.1)	9.3 (6.5, 13.4)	21.5 (14.6, 31.6)	0.069	**<0.001**	0.111
**QUICKI**	0.353 ± 0.022	0.303 ± 0.015	0.347 ± 0.021	0.305 ± 0.015	0.366	**<0.001**	0.090
**BMI, kg/m^2^**	24.8 ± 3.6	26.0 ± 3.8	24.5 ± 4.1	26.0 ± 6.9	0.791	**0.013**	0.775
**WHtR**	0.48 ± 0.05	0.48 ± 0.05	0.49 ± 0.09	0.49 ± 0.09	0.301	0.766	0.611
**TBF, %**	21.1 ± 7.1	23.5 ± 7.4	34.7 ± 7.0	35.7 ± 8.8	**<0.001**	0.066	0.373
**SBP, mmHg**	125 ± 11	126 ± 13	110 ± 11	110 ± 10	**<0.001**	0.899	0.551
**DBP, mmHg**	74 ± 7	76 ± 9	72 ± 8	73 ± 9	**0.008**	0.381	0.680
**HDL-C, mmol/L**	1.27 ± 0.23	1.16 ± 0.23	1.46 ± 0.30	1.48 ± 0.34	**<0.001**	0.129	**0.039**
**nonHDL-C, mmol/L**	2.75 ± 0.72	3.13 ± 0.91	2.79 ± 0.58	2.95 ± 0.81	0.417	**0.002**	0.220
**TAG, mmol/L**	0.85 (0.55, 1.30)	1.25 (0.75, 2.07)	0.81 (0.59, 1.12)	1.08 (0.67, 1.72)	0.060	**<0.001**	0.267
**cMSS**	2.09 ± 0.44	2.58 ± 0.74	2.00 ± 0.38	2.44 ± 0.51	**0.001**	**<0.001**	**0.044**
**cMSS4**	1.23 ± 0.43	1.64 ± 0.75	1.17 ± 0.38	1.35 ± 0.50	**0.005**	**<0.001**	0.057
**Albumin, g/L**	48.5 ± 2.0	48.2 ± 2.2	47.4 ± 2.2	46.8 ± 2.6	**<0.001**	0.074	0.566
**CRP, mg/L**	0.5 (0.2, 1.7)	0.7 (0.2, 1.9)	0.8 (0.2, 2.6)	1.4 (0.3, 5.2)	**<0.001**	**0.007**	0.232
**Uric acid, mmol/L**	360 ± 56	376 ± 65	263 ± 50	275 ± 58	**<0.001**	**0.029**	0.777
**eGFR, ml/min/1.73 m^2^**	104 ± 12	102 ± 13	107 ± 15	109 ± 14	**0.001**	0.969	0.281
**u-ACR, mg/mmoL**	0.4 (0.2, 0.9)	0.3 (0.1, 0.6)	0.5 (0.2, 1.3)	0.5 (0.2, 1.7)	**<0.001**	0.096	0.200
**TBARS, µmol/L**	1.60 (0.32, 3.16)	1.38 (0.30, 2.44)	1.17 (0.33, 2.54)	1.42 (0.30, 3.18)	0.089	0.752	**0.038**

*ANOVA* analysis of variance, *IS* insulin-sensitive (i.e., QUICKI > 0.319), IR insulin-resistant (i.e., QUICKI ≤ 0.319), Chi Chi-square test *p*, FA fructosamine, QUICKI quantitative insulin sensitivity check index, BMI body mass index; WHtR waist-to-height ratio, TBF total body fat, SBP systolic blood pressure, DBP diastolic blood pressure, HDL-C high-density lipoprotein cholesterol, TAG triacylglycerols, cMSS continuous metabolic syndrome score, cMSS4 continuous metabolic syndrome score without glycemia, CRP C-reactive protein, eGFR estimated glomerular filtration rate, u-ACR urinary albumin-to-creatinine ratio, TBARS thiobarbituric acid reactive substances. Data are presented as counts (percentage), mean ± SD (normally distributed data), or as geometric mean (−1 SD, +1 SD) of back-transformed log data (data not fitting to normal distribution were logarithmically transformed); and were evaluated using the two-factor ANOVA, with categorized sex and insulin sensitivity status entered as fix factors. *p* < 0.05 was considered significant and is highlighted in bold.

**Table 2 nutrients-14-04929-t002:** Reactive dicarbonyls, advanced glycation end-products, soluble receptors for advanced glycation end-products, and soluble vascular adhesion protein levels in sera; urinary excretion of advanced glycation end-products and D-lactate; and fractional excretion of advanced glycation end-products in insulin-sensitive and insulin-resistant males and females.

	Males (*n* = 162, 52.4%)		Females (*n* = 147, 47.6%)		Two-Factor ANOVA		
	IS (*n* = 89)	IR (*n* = 73)	IS (*n* = 78)	IR (*n* = 69)	Sex	IS/IR	Sex × IS/IR
**Methylglyoxal, nmol/L**	520 (406, 666)	512 (387, 676)	335 (256, 437)	364 (290, 457)	**<0.001**	0.244	0.089
**Glyoxal, nmol/L**	1871 ± 479	1818 ± 526	1188 ± 233	1329 ± 309	**<0.001**	0.348	**0.039**
**3-DG, nmol/L**	1492 ± 127	1666 ± 183	1128 ± 140	1305 ± 198	**<0.001**	**<0.001**	**0.004**
**free MG-H1, nmol/L**	105 (69, 160)	126 (72.4, 223)	85 (49, 146)	115 (64, 209)	**0.010**	**<0.001**	0.291
**free CML, nmol/L**	101 (74, 139)	94 (65, 134)	76 (56, 101)	78 (56, 107)	**<0.001**	0.468	0.140
**free CEL, nmol/L**	52.2 ± 16.0	57.7 ± 24.5	45.3 ± 13.8	51.6 ± 17.4	**0.002**	**0.005**	0.864
**pb-pentosidine, nmol/mmol lys**	0.39 (0.33, 0.45)	0.41 (0.35, 0.49)	0.41 (0.33, 0.51)	0.42 (0.34, 0.52)	0.061	**0.040**	0.381
**pb-MGH1, nmol/mmol lys**	320 ± 58	277 ± 42	308 ± 50	320 ± 49	**0.007**	**0.010**	**<0.001**
**pb-CML, nmol/mmol lys**	71.9 (58.5, 88.4)	66.2 (53.4, 82.0)	75.8 (60.2, 95.4)	76.2 (60.2, 95.4)	**<0.001**	0.128	0.082
**pb-CEL, nmol/mmol lys**	8.6 (6.7, 11.0)	8.2 (6.2, 10.9)	13.2 (9.0, 19.2)	18.8 (13.2, 26.9)	**<0.001**	**<0.001**	**<0.001**
**sRAGE, pg/mL**	1437 ± 470	1619 ± 573	1527 ± 495	1379 ± 421	0.184	0.765	**0.004**
**esRAGE, pg/mL**	333 ± 157	334 ± 169	308 ± 113	281 ± 101	**0.014**	0.434	0.384
**cRAGE, pg/mL**	1104 ± 363	1284 ± 441	1219 ± 406	1097 ± 357	0.435	0.514	**0.001**
**cRAGE/esRAGE**	3.50 (2.32, 5.27)	4.08 (2.96, 5.63)	4.02 (3.09, 5.22)	3.98 (2.91, 5.42)	0.142	0.063	**0.033**
**sVAP-1, ng/mL**	322 (246, 421)	349 (231, 529)	342 (242, 483)	387 (217, 688)	0.082	**0.027**	0.660
	*n* = 87	*n* = 72	*n* = 70	*n* = 61			
**u-MGH1, nmol/mmol crea**	2050 (1269, 3310)	2290 (1417, 3703)	2320 (1257, 4284)	3006 (1609, 5616)	**0.002**	**0.005**	0.255
**u-CML, nmol/mmol crea**	1028 (763, 1384)	1115 (754, 1648)	1153 (745, 1784)	1246 (791, 1965)	**0.016**	0.086	0.973
**u-CEL, nmol/mmol crea**	507 (369, 697)	536 (379, 759)	560 (401, 777)	641 (458, 897)	**<0.001**	**0.016**	0.312
**FE_MGH-1_**	1.50 (1.07, 2.10)	1.40 (1.01, 1.93)	1.63 (1.06, 2.51)	1.58 (1.01, 2.49)	**0.023**	0.258	0.628
**FE_CML_**	0.78 (0.54, 1.13)	0.93 (0.71, 1.21)	0.90 (0.67, 1.20)	0.97 (0.66, 1.42)	**0.010**	**0.002**	0.235
**FE_CEL_**	0.81 (0.66, 1.01)	0.77 (0.69, 0.94)	0.77 (0.59, 1.01)	0.78 (0.59, 1.02)	0.787	0.819	0.573
	*n* = 62	*n* = 48	*n* = 52	*n* = 40			
**u-D-lactate, µmol/mmol crea**	4.2 (1.8, 9.6)	3.6 (1.7, 7.5)	6.3 (3.0, 13.0)	7.1 (3.7, 13.4)	**<0.001**	0.948	0.166

*ANOVA* analysis of variance, *IS* insulin-sensitive (i.e., QUICKI > 0.319), *IR* insulin-resistant (i.e., QUICKI ≤ 0.319), *3-DG* 3-deoxyglucosone, *MG-H1* methylglyoxal-derived hydroimidazolone, *CML* N^ε^-(carboxymethyl)lysine, *CEL* N^ε^-(carboxyethyl)lysine, *pb* protein bound, *lys* lysine, *sRAGE* soluble. receptor for advanced glycation end-products, *esRAGE* endogenous secretory receptor for advanced glycation end-products, *cRAGE* cleaved soluble receptor for advanced glycation end-products, *sVAP-1* soluble vascular adhesion protein-1, *u* urinary, *crea* creatinine, *FE* fractional excretion. Data are presented as counts (percentage), mean ± SD (normally distributed data), or as geometric mean (−1 SD, +1 SD) of back-transformed log data (data not fitting to normal distribution were logarithmically transformed); and were evaluated using the two-factor ANOVA, with categorized sex and insulin sensitivity status entered as fix factors. *p* < 0.05 was considered significant and is highlighted in bold.

**Table 3 nutrients-14-04929-t003:** Simple regression (Pearson correlation coefficient) and multiple regression using the orthogonal projections to latent structures (OPLS) model of independent variables on the quantitative insulin sensitivity check index (QUICKI) in males and females.

	Pearson Correlation				OPLS (VIP)	
	Males (*n* = 162)	*p*	Females (*n* = 147)	*p*	Males	Females
**CMSS4**	−0.308	**<0.001**	−0.359	**<0.001**	**1.44**	**1.39**
**Fructosamine**	−0.259	**0.001**	−0.070	0.401	**1.22**	0.29
**Total body fat**	−0.179	**<0.001**	−0.255	0.002	**1.15**	**1.09**
**Non-HDL-C**	−0.204	**0.009**	−0.154	0.062	**1.12**	0.85
**Log C-reactive protein**	−0.187	**0.024**	−0.198	**0.016**	0.69	0.94
**Uric acid**	−0.075	0.345	−0.120	0.147	NI	NI
**EGFR**	−0.032	0.690	−0.010	0.224	NI	NI
**Log ACR**	0.162	0.059	−0.078	0.383	NI	NI
**Log TBARS**	0.095	0.233	−0.163	0.051	0.77	0.76
**Log methylglyoxal**	−0.015	0.849	−0.208	**0.011**	0.90	0.98
**Glyoxal**	0.026	0.741	−0.198	**0.016**	0.92	**1.08**
**3-DG**	−0.553	**<0.001**	−0.546	**<0.001**	**1.86**	**1.99**
**Log free MG-H1**	−0.165	**0.036**	−0.310	**<0.001**	0.73	**1.18**
**Log free CML**	0.115	0.144	−0.064	0.443	0.73	0.96
**Free CEL**	−0.110	0.163	−0.165	**0.045**	0.71	**1.11**
**Log pb-pentosidine**	−0.066	0.403	0.059	0.479	0.57	0.19
**Pb-MGH1**	0.354	**<0.001**	0.012	0.884	**1.50**	0.13
**Log pb-CML**	0.167	**0.033**	0.111	0.181	0.90	0.58
**Log pb-CEL**	0.115	0.144	−0.348	**<0.001**	0.87	**1.20**
**SRAGE,**	−0.146	0.064	0.165	**0.046**	NI	NI
**EsRAGE**	−0.015	0.853	0.067	0.418	NI	NI
**CRAGE,**	−0.182	**0.021**	0.179	**0.030**	**1.04**	0.77
**Log cRAGE/esRAGE**	−0.153	0.051	0.123	0.137	NI	NI
**Log sVAP-1**	−0.140	0.075	−0.073	0.378	0.76	0.39
**Log u-MGH1**	−0.104	0.191	−0.218	**0.012**	0.73	**1.23**
**Log u-CML**	−0.092	0.248	−0.122	0.166	0.58	0.97
**Log u-CEL**	−0.090	0.257	−0.196	**0.025**	0.66	0.99
**Log FE_MGH-1_**	0.180	0.176	0.040	0.651	NI	NI
**Log FE_CML_**	−0.204	**0.010**	−0.082	0.353	NI	NI
**Log FE_CEL_**	0.049	0.536	−0.063	0.472	NI	NI
	*n* = 110		*n* = 92			
**Log u-D-lactate**	0.090	0.349	−0.192	0.066	NI	NI
**R^2^**	-	-	-	-	0.46	0.48

*cMSS4* continuous metabolic syndrome score without glycemia, *HDL-C* high-density lipoprotein cholesterol, *NI* not included, *eGFR* estimated glomerular filtration rate, *ACR* urinary albumin-to-creatinine ratio, *TBARS* thiobarbituric acid reactive substances, *3-DG* 3-deoxyglucosone, *MG-H1* methylglyoxal-derived hydroimidazolone, *CML* N^ε^-(carboxymethyl)lysine, *CEL* N^ε^-(carboxyethyl)lysine, *pb* protein bound, *sRAGE* soluble receptor for advanced glycation end-products, *esRAGE* endogenous secretory receptor for advanced glycation end-products, *cRAGE* cleaved soluble receptor for advanced glycation end-products, *sVAP-1* soluble vascular adhesion protein-1, *u* urinary, *FE* fractional excretion, significant *p* for simple correlations and variable importance for the projection values (VIP > 1.0) considered as significant predictors in the OPLS regression models are given in bold.

## Data Availability

Data are available on request from the corresponding author.

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
