# Peer review of "Association of α-Dicarbonyls and Advanced Glycation End Products with Insulin Resistance in Non-Diabetic Young Subjects: A Case-Control Study"

_nutrients, 2022, doi:10.3390/nu14224929_

Round 1

Reviewer 1 Report

In this paper the authors highlights a possible role of Dicarbonyls and advanced glycation end products (AGEs) in insulin resistance pathogenesis. However, I have several comments for the authors.

1) the paper is a cross sectional study; thus, you could consider an association between these variables and insulin resistance. Please modifiy the sentences that this study clarifies the role of these variables in the insulin resistance pathogenesis.

2) I don't understand what is the specific aim of the study? moreover, please insert the sample size.

3) Previous studies focused on the role of AGE and sRAGE and RAGE ligands in the cardiovascular risk of prediabetic subjects; please consider these studies (10.3390/cells8080910; 10.1007/s00592-019-01338-1) and comment these in the paper.

Author Response

We would like to thank the reviewer for her/his time spent evaluating our manuscript and for stimulating remarks.

In this paper, the authors highlights a possible role of Dicarbonyls and advanced glycation end products (AGEs) in insulin resistance pathogenesis. However, I have several comments for the authors.

  • the paper is a cross sectional study; thus, you could consider an association between these variables and insulin resistance. Please modifiy the sentences that this study clarifies the role of these variables in the insulin resistance pathogenesis.

As suggested by the reviewer, in the revised paper, we mention the potential role of the studied variables in the pathogenesis of insulin resistance (lines 300-2). We are cautious in our statement, as the observed associations may not unequivocally point to a pathogenetic role - the association is not causation.

  • I don't understand what is the specific aim of the study? moreover, please insert the sample size.

As suggested by the reviewer, we have modified the introduction so that the aim of the study is clear (lines 72-3). The sample size was given in the original manuscript in the abstract, methods section, and tables.

3) Previous studies focused on the role of AGE and sRAGE and RAGE ligands in the cardiovascular risk of prediabetic subjects; please consider these studies (10.3390/cells8080910; 10.1007/s00592-019-01338-1) and comment these in the paper.

We are aware of the papers of the group from Bari. As to the lower levels of esRAGE in IR adults, we cited the Bari group in the submitted paper (Ref. No.: 19). However, as suggested by the reviewer, we included also the more recent reference (2nd added citation in the revised paper). We suppose that the small, albeit well-controlled, study on S100A12, sRAGE, esRAGE, and PWV in hypercholesterolemic adults with or without a genetically confirmed diagnosis of familial hypercholesterolemia, advised for consideration by the reviewer, is not tackling issues investigated in our study.

Reviewer 2 Report

1). Explain the arbitrary cut off for QUICKI, why 15th percentile, i.e., QUICKI ≤ 0.319 was chosen, not 20 or 10%.

2). Need to explicitly specify the outcome (dependent variable, assuming QUICKI here) and input (independent variables).

3). Claim of independent variables for the multiple parameters in the regression model is likely problematic. For example, cmss 4, body fat, non-HDL-C... etc (Table 3). They are likely not independent. This is probably the stickiest points that may invalidate the results to some degree (may not change the conclusion, though). Multicollinearity issue.  Maybe ridge regression, principal component regression, or partial least squares regression be better choice? 

Author Response

We would like to thank the reviewer for her/his time spent evaluating of our manuscript and for stimulating remarks.

1). Explain the arbitrary cut off for QUICKI, why 15th percentile, i.e., QUICKI ≤ 0.319 was chosen, not 20 or 10%.

There neither are recommendations from professional societies regarding the threshold values of surrogate markers to define insulin resistance, nor a consensus on the cutoff point’s determination based on the percentile criterion. Percentile criterions employed by different groups range profoundly, as they assign 10%-to-33% of the studied population as IR (doi: 10.1186/1472-6823-13-47). Most frequently, 10% or 25% of the studied populations are arbitrarily assigned as IR. Based on the arbitrariness of the decision, we selected the 15th percentile of the QUICKI, as a (downward rounded) compromise between the 2 most frequent arbitrary classifications. We inserted an explanation in the revised paper (lines 93-5).

2). Need to explicitly specify the outcome (dependent variable, assuming QUICKI here) and input (independent variables).

As suggested, the sentence has been reformulated (lines 170-2).

3). Claim of independent variables for the multiple parameters in the regression model is likely problematic. For example, cmss 4, body fat, non-HDL-C... etc (Table 3). They are likely not independent. This is probably the stickiest points that may invalidate the results to some degree (may not change the conclusion, though). Multicollinearity issue.  Maybe ridge regression, principal component regression, or partial least squares regression be better choice? 

Many thanks for this remark. We agree with the reviewer that metabolic markers of increased cardiometabolic risk are logically metabolically interrelated. Albeit OPLS is robust (among others) against partially intercorrelated data, and thus a commonly used way to correct multicollinearity, we always check for multicollinearity between considered independent variables. Variance inflation factors for QUICKI as a dependent factor and CMSS4, TBF, non-HDL-C, fructosamine, cRAGE, log CRP, log TBARS, and log sVAP-1 as predictors varied between 1.07 (log sVAP-1) and 2.13 (cMMS4) in males and 1.06 (log TBARS) and 1.61 (cMSS4) in females. Thus, correlations between given predictor variables are moderate. In the revised paper, we mention that testing for multicollinearity has been performed prior to multivariate regression (lines 168-170).

Round 2

Reviewer 1 Report

The authors satisfied the requested revision